# Optimization of Hyperparameters in Object Detection Models Based on Fractal Loss Function

**Ming Zhou** , **Bo Li** * **and Jue Wang** *

School of Information Science and Engineering, Dalian Polytechnic University, Dalian 116039, China
* Correspondence: libolb@dlpu.edu.cn (B.L.); wangjue@dlpu.edu.cn (J.W.)

**Abstract:** Hyperparameters involved in neural networks (NNs) have a significant impact on the accuracy of model predictions. However, the values of the hyperparameters need to be manually preset, and finding the best hyperparameters has always puzzled researchers. In order to improve the accuracy and speed of target recognition by a neural network, an improved genetic algorithm is proposed to optimize the hyperparameters of the network by taking the loss function as the research object. Firstly, the role of all loss functions in object detection is analyzed, and a mathematical model is established according to the relationship between loss functions and hyperparameters. Secondly, an improved genetic algorithm is proposed, and the feasibility of the improved algorithm is verified by using complex fractal function and fractional calculus. Finally, the improved genetic algorithm is used to optimize the hyperparameters of the neural network, and the prediction accuracy of the model before and after the improvement is comprehensively analyzed. By comparing with state-of-the-art object detectors, our proposed method achieves the highest prediction accuracy in object detection. Based on an average accuracy rate of 95%, the detection speed is 20 frames per second, which shows the rationality and feasibility of the optimized model.

**Keywords:** neural networks; genetic algorithm; target detection





## 1. Introduction

Object detection is an applied mathematical technique based on the geometric and statistical characteristics of objects [1]. Physical properties related to various physical phenomena based on technologies also play a key role in object detection, such as electromagnetics, acoustics, optics with scattering, emission, and absorption. In recent years, with the continuous development of science and technology, target detection has played an important role in various fields, such as intelligent monitoring, medical navigation surgery, military target detection, etc. However, many models fail when faced with various complex scenarios and real-time processing of targets [2]. In order to improve the ability of computer vision to cope with complex environments, various scholars have made efforts [3]. The research trends for improving performance can be divided into two directions: one is to improve the ability of model feature extraction from the network framework, such as geometric detectors, attention mechanisms, pyramid networks, etc.; the other is to start from the loss function used to optimize hyperparameters in the network, such as Bayesian-based hyperparameter optimization and whale optimization algorithm-based hyperparameter optimization.

Taking into account the impact of the backbone network on object detection performance. Hua et al. [4] proposed a matrix-information geometry detector based on Bregman divergence. The author first establishes a positive definite matrix and a clutter covariance matrix for each sample and then redefines the points on the matrix manifold as the discriminator for signal detection. The final experimental results show that the proposed model is stronger than other detectors. Dai et al. [5] propose a model-driven network for small object detection, where the authors transform traditional local contrast measurements into deep

unparameterized feature refinement layers. In addition, the authors design a bottom-up attention mechanism to transform subtle details into higher-level feature maps. Through ablation experiments, it is proved that the accuracy of the changed network is significantly improved. Zhan et al. [6] proposed a feature pyramid network based on parallel spatial domain attention mechanism and small-scale transformers. The author fully extracts the texture of the target, which effectively improves the extraction ability of small targets. Validated on a home-made dataset, the results show that the proposed network has better performance than previous methods. However, these methods are only improved for the network framework, and they do not seem to realize the importance of parameters. It is understood that the setting of hyperparameters has a direct impact on the performance of the model, not only in terms of detection accuracy and training speed [7]. Therefore, some scholars propose to use intelligent optimization algorithms to optimize hyperparameters.

Considering the impact of hyperparameters on object detection performance, a large number of scholars optimize these hyperparameters using intelligent optimization algorithms. With the loss function as the objective function, the set of hyperparameter values with the smallest loss will be given as the optimal hyperparameter. Here, we emphasize the role of the loss function. Generally speaking, the loss function is used to calculate the gap between the forward calculation result of each iteration of the neural network and the real value, so as to guide the next step of training in the right direction. Victoria et al. [8] propose a model based on Bayesian optimization of hyperparameters. The authors validate the performance of the Bayesian hyperparameter optimization algorithm on the CIFAR-10 dataset. The results show that the Bayesian optimization algorithm model saves time and improves performance during the training phase. Brodzicki et al. [9] proposed a whale-based optimization algorithm to optimize hyperparameters. The authors highlight the difficulty of the Whale algorithm in the hyperparameter optimization task and compare it with other state-of-the-art algorithms. By searching for objects in 3D space, the results show that the proposed algorithm achieves an average accuracy of up to 85%. Lee et al. [10] proposed a neural network structure and hyperparameter optimization method based on genetic algorithm. The authors highlight the impact of different hyperparameters on the convergence of convolutional neural network (CNN) models, showing the possibility of Genetic Algorithm (GA) optimizing the network framework. Validated on a self-made image dataset, the results show that the proposed algorithm outperforms the equivalent object detection algorithm by 11.73%.

However, in the above-mentioned published papers, the existing optimization algorithms are only directly applied to target detection, which obviously cannot meet the practical challenges [11]. Furthermore, they simply optimize the initial hyperparameters. It is understood that the hyperparameters required in the object detection task should be updated in real time to improve the effect of model detection [12]. Therefore, this paper proposes a deep learning hyperparameter optimization framework based on an improved genetic algorithm. The main contributions of this paper are as follows:

(1) An improved genetic algorithm is proposed to solve the problem of objective optimization;
(2) The improved genetic algorithm is proposed to optimize the hyperparameters of the neural network;
(3) Determine a reasonable fitness function according to the relationship between the loss function and hyperparameters, and establish a mathematical model;
(4) The superiority of the proposed method in the task of object detection is demonstrated by comparing with state-of-the-art object detection algorithms.

The rest of this article is organized this way. Section 2 briefly introduces the classification loss function and regression loss function in target detection and expounds the improved genetic algorithm. Section 3 introduces the improved genetic neural network and establishes the mathematical model according to the actual problem. Section 4 develops qualitative and quantitative analyses to demonstrate the superiority of the proposed method. Section 5 is the conclusion.

## 2. Related Work

### 2.1. Loss Function for Object Detection

The loss function is used to evaluate the degree of difference between the predicted value of the model and the true value [13]. Specific to target detection, the function of the loss function is to make the recognition accuracy higher and the positioning more accurate. The loss function in the object detection task consists of a classification loss (Lcls) and a regression loss (Lbox) [14]. The optimization process of the classification problem is essentially the process of minimizing the loss function. In the task of multi-classification, the softmax function is usually used. It maps the outputs of multiple neurons to the (0, 1) interval, and selects the largest value as the final predicted category according to the probability value output by each neuron. It is worth emphasizing that in the process of prediction, there is a competitive relationship between different categories [15]. Ideally, the model predicts a class with the highest probability value and the rest are very low. The labels of objects are predicted according to a dedicated classification branch, each label representing a different class. The loss function used for the classification task is shown in Equation (1).

$$L_{cls} = loss_1(P_i, class) = -\log\left(\frac{\exp(P[class])}{\sum_j \exp(x[j])}\right) = -P[class] + \log\left(\sum_j \exp(x[j])\right) \quad (1)$$

Among them, $loss_1$ is the defined first-class loss function, which has two parameters ($P_i$ and class), $P_i$ represents the probability predicted by the model when dealing with multi-classification problems, $i$ represents the labels of all targets in the multi-classification task, and class represents the class assigned to each codes [0, 1, 2 . . . ] of classes, and $j$ represents the total number of all classified objects.

Logistic regression is a supervised classification model that is often used to describe the difference between predictions and reality [16]. Therefore, the regression loss function is used to evaluate the size of the information loss caused by the model fitting training. The smaller the loss function, the better the model fits on the training set. The loss function for logistic regression is shown in Equation (2). The regression of bounding boxes in SSD and Faster RCNN uses Smooth L1 as the iterative function.

$$loss_2\left(F_{nj}^*, G_{nj}\right) = smooth_{L1}\left(F_{nj}^*, G_{nj}\right) = \begin{cases} 0.5\left(F_{nj}^* - G_{nj}\right)^2, & \left|F_{nj}^* - G_{nj}\right| < 1 \\ \left|F_{nj}^* - G_{nj}\right| - 0.5, & \left|F_{nj}^* - G_{nj}\right| \geq 1 \end{cases} \quad (2)$$

$$L_{reg} = \frac{1}{N} \sum_{n=1}^{N} t_n^* \sum_{j \in \{x,y,w,h\}} loss_2\left(F_{nj}^*, G_{nj}\right) \quad (3)$$

Among them, $F_{nj}^*$ represents the predicted value, $G_{nj}$ represents the real value, $N$ represents the total number of anchors (here $N = 9$), and $t_n^*$ represents the encoding of the area within the target frame (the background is marked with 0, and the target area is marked with 1).

### 2.2. Intelligent Optimization Algorithm

Optimization problem is an applied technique based on mathematics for solving various optimization problems [17]. It is widely used in image processing, pattern recognition, automatic control and signal processing and other fields. A common optimization problem refers to finding the optimal parameter values and solutions among many parameters and solutions under given conditions so that multiple performance indicators are optimal. Inspired by biological groups or the laws of natural development, many intelligent optimization algorithms are used to solve practical engineering problems [18]. For example, genetic algorithms that imitate biological evolution mechanisms; differential evolution algo-

rithms that optimize search through cooperation and competition among individuals; ant colony algorithms that simulate collective path-finding behavior of ants. These algorithms have in common that they were developed by simulating or revealing certain natural phenomena and processes [19]. They can be roughly divided into three categories: evolutionary algorithms, swarm intelligence algorithms, and simulated annealing algorithms.

Evolutionary computing is a series of search technologies, including genetic algorithms, evolutionary algorithms, etc., which are widely used in machine learning, neural network training, intelligent control and other fields. Among them, the genetic algorithm is the most commonly used representative. In [20], a control strategy method based on improved genetic algorithm is proposed. The authors emphasize the role of artificial intelligence in promoting social development, and apply the proposed algorithm to the governance of living standards, solving practical problems, and assessing epidemic prevention and control. The research results show that the problem solving rate increased by more than 50% after the introduction of the improved genetic algorithm. The immune algorithm is an intelligent search algorithm constructed by imitating the biological immune mechanism and combining the genetic evolution mechanism. Compared with other algorithms, the immune algorithm has its own production diversity and maintenance mechanism, which avoids the "premature" problem and can obtain the global optimal solution.

Swarm intelligence is a computing technology based on the behavioral laws of biological groups. Most global optimization problems can be solved efficiently with centralized control and no global model. At present, there are two main algorithms in the research field of swarm intelligence theory: ant colony algorithm and particle swarm algorithm. The former is to simulate the food collection process of the ant colony, and the latter is to simulate a simple social system. Liang et al. [21] proposed an improved ant colony optimization algorithm. The author modeled according to the weather, comfort and travel route of the scenic spot, and introduced sub-road support to avoid falling into local optimum. The experimental results show that the optimized route greatly improves the travel experience.

The simulated annealing algorithm selects a state with a large target value in the field with a certain probability and has a very strong global search performance. It uses probabilistic transitions to guide its search direction, and these probabilities are just a tool to guide its search process toward a region of more optimal solutions. Ilhan et al. [22] proposed an improved simulated annealing algorithm, and the author proposed the crossover operator of ISA-CO for the first time in the paper. Partial map crossover and order crossover operators are applied to the in-swarm solution to speed up convergence, and a hybrid selection method is used to ensure a balance between exploitation and search. The results show that this method in most cases outperforms other state-of-the-art methods.

## 3. The Proposed Methods

### 3.1. Improved Genetic Algorithm

Genetic Algorithm (GA) is a computational model that simulates the genetic mechanism and the biological evolution process of natural selection [23]. Relying on its excellent global search ability, genetic algorithms are widely used in engineering applications. However, blindly applying GA directly to real projects can lead to many problems. For example, it is prone to premature phenomenon, it is difficult to obtain the global optimum, and the optimization speed is slow and detours. In this paper, an improved genetic algorithm is proposed to solve the problems of low precision and slow speed in target detection.

Evolutionary algorithms were developed by drawing on phenomena, such as inheritance, mutation, natural selection, and mutation. The process of natural selection is based on fitness evaluation, and an unreasonable fitness function will lead to convergence to a local optimum [24]. In this paper, we use Equation (4) as the fitness function, which can well reflect the performance of the model during the training phase. In addition, elite retention strategy and roulette, as the most classic selection operators, have their own advantages and disadvantages [25]. Specifically, elite retention is to directly copy the optimal individuals that appear in each round to the next generation, without participating in selection,

crossover and mutation operations to retain samples. However, it ignores the diversity of species and tends to limit the results to local optima. The roulette approach focuses on species diversity but cannot guarantee the survival of optimal samples [26]. Therefore, in order to get a better selection operator, we learn from the idea of elite reservation, and directly copy the 5% individuals with higher fitness to the next generation. In addition, 10% individuals with higher fitness will replace individuals with lower fitness, and the newly formed community will complete the process of crossover and mutation and inherit it to the next generation (as shown in Figure 1).

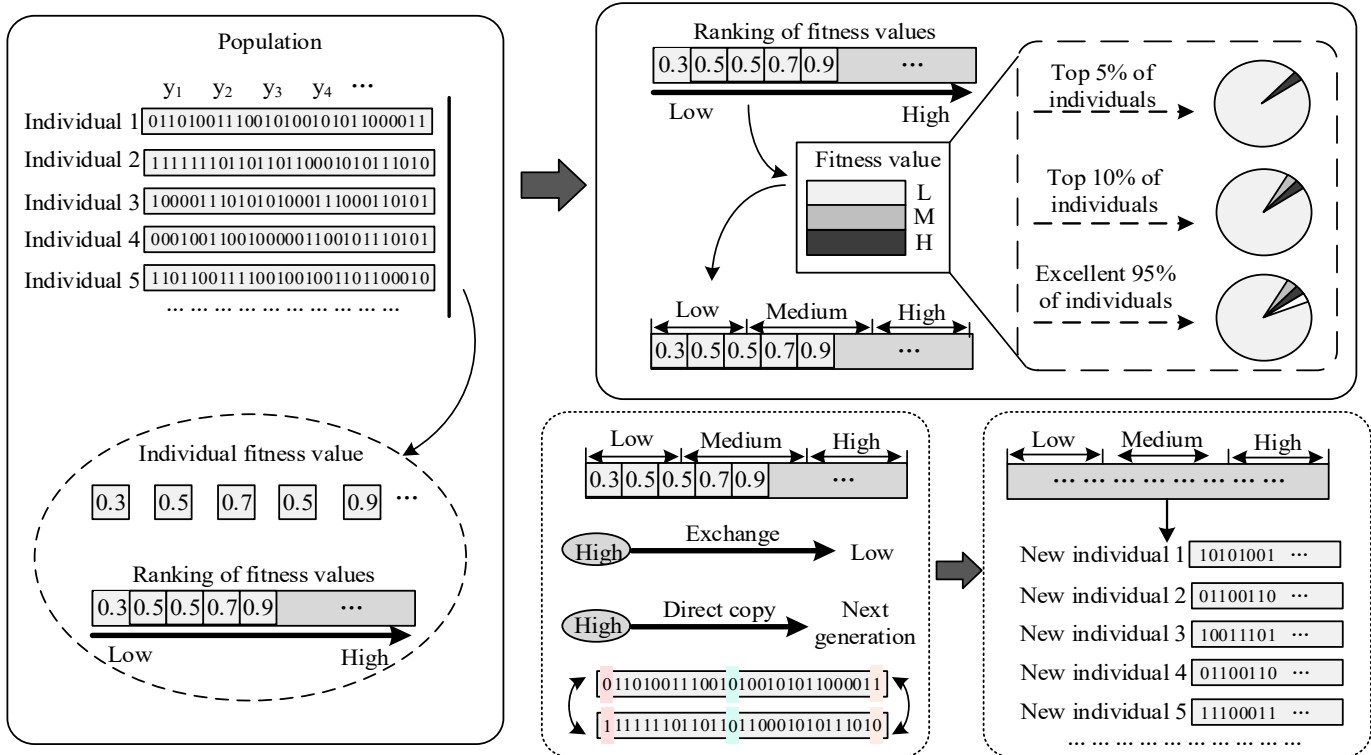

**Figure 1.** The proposed improved genetic algorithm.

However, the above process leads to an increase in the number of populations as the number of evolutions increases, which obviously increases the computational load of the network [27]. In order to reduce the amount of computation, we sorted the fitness of all individuals, and inherited the top 95% of the individuals to the next generation, while the rest were directly discarded. This method is not good enough. Although the optimal individuals participate in the crossover and mutation operations, it is easy to fall into a local optimum by directly copying the top 5% of the optimal individuals to the next generation each time. In order to obtain the global final solution, we increased the mutation rate to Pm = 0.1 based on the original Pm = 0.03.

### 3.2. Improved Genetic Neural Network

Hyperparameters in a neural network have a crucial impact on the training effect of the model [28]. Random selection of hyperparameters will cause the model to easily fall into problems such as local minimum and poor search ability. Using the global search ability of the genetic algorithm, more reasonable hyperparameters can be obtained. The optimization objects of the genetic algorithm are mainly the learning rate, the number of training rounds, and the number of hidden layers, which make the neural network have the ability of self-adaptation and self-evolution [29,30]. Using the loss function obtained in each round of training as the reference object, the hyperparameters are modified. It is worth emphasizing that this is a dynamic process. Specifically, if the current number of training

rounds is 200, but the loss function will no longer decrease after the 100th round, the model will consider ending work after approximately 110 rounds. The improved genetic neural network is shown in the Figure 2 below.

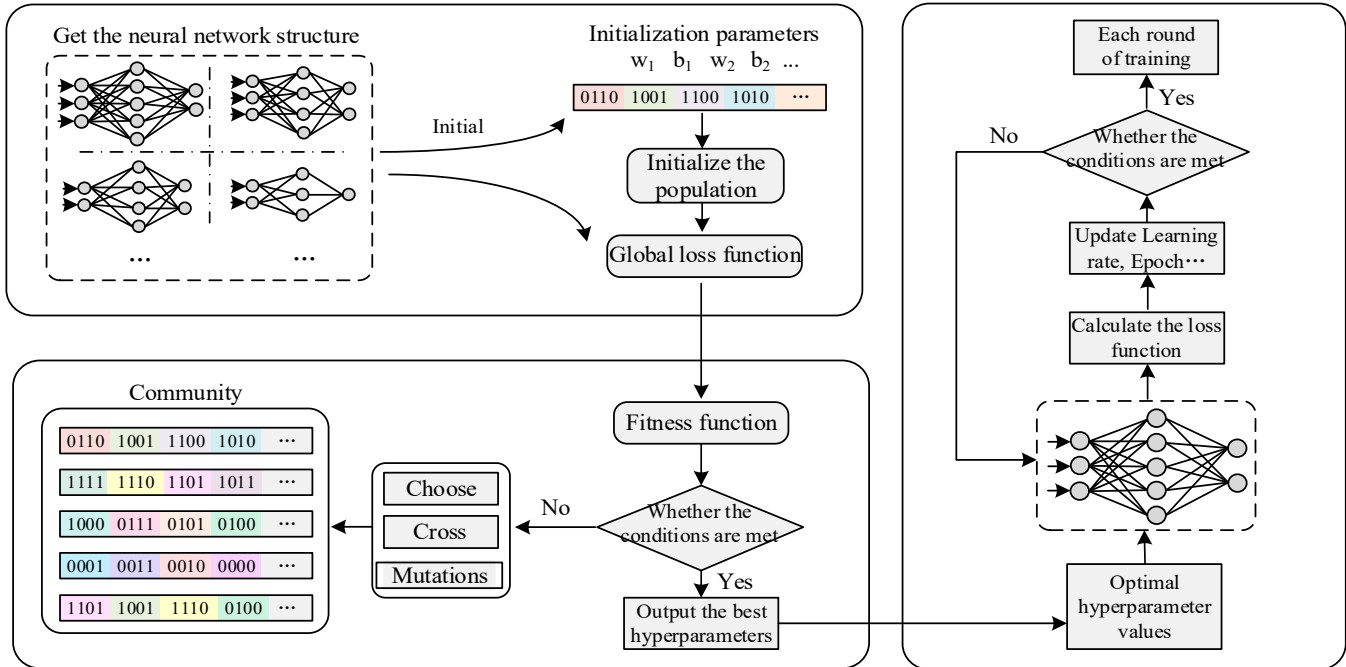

**Figure 2.** The proposed improved genetic neural network.

The improved genetic neural network consists of eight parts. First, build the basic structure of the neural network according to the actual problem. For example, when performing a multi-classification task, the number of output nodes must be the same as the type of classification. Second, initialize the population and encode the hyperparameters (learning rate, number of iterations, number of training rounds, number of hidden layer units, number of hidden layer layers) for each individual of the population [31,32]. Third, calculate the fitness value of each individual through the loss function and fitness function, and sort them according to the size of the fitness value, and select the optimal top 5% and the top 95%. Fourth, the individuals of the newly built community are randomly matched into pairs, each individual is crossed according to a certain crossover rate, and each individual in the population is subjected to mutation operation with a probability of 0.1. Fifth, the fitness of individuals in the new population is calculated. If the termination condition is met, the update of the community is stopped, otherwise, re-selection, crossover and mutation operations are performed. Sixth, select the optimal fitness and complete the decoding to obtain the optimal solution. Seventh, the optimal hyperparameters are substituted into the neural network to complete the update of weights and biases. Eighth, when the loss function changes abnormally during training, these hyperparameters will be redefined. Under normal circumstances, the learning rate will be larger in the early stage of training, and the change of the loss function after n rounds of training becomes small, so the learning rate can be adjusted appropriately [33].

### 3.3. Fractal Dimension Calculation Method

Consider that the classification loss is often directly related to how well the model predicts. If the classification loss is too large, it indicates that the model cannot accurately extract the features of the image. It even leads to confusion among similar categories, which is a fatal effect in the object recognition process [34]. However, most of the existing algorithms fail to satisfy the more precise definition of loss function. To solve the above problem, we redefine the total loss function. We divide the loss function into a regression

box loss function, a confidence loss function, and a classification loss function. The overall loss function is shown in Equation (4).

$$
\begin{aligned}
f(x) &= loss(object) \\
&= \lambda_{coord} \sum_{i=0}^{K \times K} \sum_{j=0}^{M} I_{ij}^{obj} (2 - w_i \times h_i) \left[ (x_i - \hat{x}_i)^2 + (y_i - \hat{y}_i)^2 \right] \\
&+ \lambda_{coord} \sum_{i=0}^{K \times K} \sum_{j=0}^{M} I_{ij}^{obj} (2 - w_i \times h_i) \left[ (w_i - \hat{w}_i)^2 + \left( h_i - \hat{h} \right)^2 \right] \\
&+ \lambda_{obj} \sum_{i=0}^{K \times K} \sum_{j=0}^{M} I_{ij}^{obj} \left[ \hat{C}_i \log(C_i) + \left( 1 - \hat{C}_i \right) \log(1 - C_i) \right] \\
&+ \lambda_{noobj} \sum_{i=0}^{K \times K} \sum_{j=0}^{M} I_{ij}^{noobj} \left[ \hat{C}_i \log(C_i) + \left( 1 - \hat{C}_i \right) \log(1 - C_i) \right] \\
&+ \lambda_{obj} \sum_{i=0}^{K \times K} I_{ij}^{obj} \sum_{c \in classes} \left[ \hat{p}_i(c) \log(p_i(c)) + (1 - \hat{p}_i(c)) \log(1 - p_i(c)) \right]
\end{aligned}
\tag{4}
$$

Here, *coord* indicates that the input layer has additional co-ordinate information channels, *P* indicates the predicted classification probability, *i* and *j* indicate co-ordinate information, $(x, y)$ indicates the center co-ordinates of the rectangular box, w and h indicate the width and height of the rectangular box, *C* stands for confidence, *M* stands for the number of categories, *K* stands for all clusters, $\lambda$ stands for the parameters used to balance these loss functions, *c* stands for the class of classification, and those symbols with crowns represent the true value.

As a loss function to evaluate the gap between prediction and reality, it needs to be judged by multi-angle information, and these observation angles also have self-similarity. Considering the multifractal nature of the loss function, we analyze and evolve a system consisting of many interacting elements in space and time. First, assuming that the entire agent population is distributed in a regular space, a representation of the spatial co-ordinate $(x, y)$ at time t is sought. Time t is used to describe the number of training rounds, and $(X, Y)$ is used to describe the relevant data of the predicted value. In this paper, the box counting method is used to estimate the fractal dimension of an object as the slope of a linear fit between the logarithm of the number of boxes required to optimally cover the object and the logarithm of the box size. We follow Wang et al. [35] and consider defining the fractal loss function in terms of the integral and its continuous limit time derivative, as shown in the following equation.

$$
\iint R(x,y) \frac{\partial^\beta f(x,y,t)}{\partial t^\beta} dxdy = \lim_{\Delta t \to 0} \iint R(x,y) \frac{f(x,y,t+\Delta t) - f(x,y,t)}{\Delta t^\beta} dxdy
\tag{5}
$$

Here, *x* represents the center co-ordinates of the prediction frame, y represents the size of the prediction frame, *R* represents an arbitrary function (generally refers to weighing the importance of each parameter by establishing weights), and *t* represents time (number of rounds). Assume that the evolution of $f(x,y,t)$ is fractal in time, and express the fractal by fractional derivatives of order $\beta$. Using the Bayesian variation, the expression in Equation (6) is updated as:

$$
\begin{aligned}
&\lim_{\Delta t \to 0} \iint R(x,y) \frac{f(x,y,t+\Delta t) - f(x,y,t)}{\Delta t^\beta} dxdy = \\
&\lim_{\Delta t \to 0} \iint \frac{R(x,y)}{\Delta t^\beta} \left\{ \iint f\left( x,y,t + \Delta t \overrightarrow{AB} \mid_{x_1,y_1,t} \right) . f(x_1,y_1,t) dx_1 dy_1 - f(x,y,t) \right\} dxdy
\end{aligned}
\tag{6}
$$

Reversing the order of integration and using the fractional Taylor expansion $R(x,y)$ is expressed as:

$$
\begin{aligned}
R(x,y) &= R(x_1,y_1) + (x-x_1)^\gamma \frac{\partial^\gamma R(x,y)}{\partial x^\gamma}\bigg|_{\substack{(x,y)=\\(x_1,y_1)}} \\
&+ (y-y_1)^\gamma \frac{\partial^\gamma R(x,y)}{\partial y^\gamma}\bigg|_{\substack{(x,y)=\\(x_1,y_1)}} + (x-x_1)^{2\gamma}\frac{\partial^{2\gamma} R(x,y)}{\partial x^{2\gamma},y_1)}\bigg|_{\substack{(x,y)=\\(x_1,y_1)}} \\
&+ (x-x_1)^\gamma (y-y_1)^\gamma \left[\frac{\partial^{2\gamma} R(x,y)}{\partial x^\gamma \partial y^\gamma}+\frac{\partial^{2\gamma} R(x,y)}{\partial y^\gamma \partial x^\gamma}\right]_{\substack{(x,y)=\\(x_1,y_1)}} \\
&+ (y-y_1)^{2\gamma}\frac{\partial^{2\gamma} R(x,y)}{\partial y^{2\gamma}}\bigg|_{\substack{(x,y)=\\(x_1,y_1)}} + O\left(x^{2\gamma},y^{2\gamma}\right) + \dots
\end{aligned}
\tag{7}
$$

where $\gamma$ is the fractional order related to the fractal structure of agent evolution, and $(x, y)$ represents the center co-ordinates of the training box in the first round.

### 3.4. The Proposed Genetic Neural Network

The initialization of the population is essentially to give the initial solution of the population according to the coding rules [36]. Population initialization is the first and most important step of the algorithm. Common initializations include fixed value setting method, M-type random method and two-step method. The fixed value setting method is more dependent on the range of feasible solutions, and is often used to search for uniformly distributed points in the space. When faced with large-scale optimization problems, random generation methods are easily limited to local optimal solutions. The two-step method is by far the most commonly used method. It is divided into early stage and late stage. The early stage is generally generated randomly, and the later stage is adjusted according to the change of the fitness function (Equation (4)). Considering the complexity of the change of the loss function in the target recognition task, this paper creatively proposes a population initialization method based on the global loss mixed sorting [37]. The design concept of this method is to infer a set of optimal solution candidates as the initial population through the loss function mixture matrix.

a　The initialization of the population is provided by the following Equation (5).

$$
HM = \begin{bmatrix}
N^{ck}_{11} & S^{ck}_{21} & N^{nl}_{11} & F^{ac}_{2i} & R^{lea}_{x1} & N^{bs}_{y1} & W^{dc}_{w1} & R^{dro}_{h1} & \hat{G}_{\theta 1} \\
\vdots & \vdots & \vdots & \vdots & \vdots & \vdots & \vdots & \vdots & \vdots \\
N^{ck}_{1i} & S^{ck}_{2i} & N^{nl}_{1i} & F^{ac}_{2i} & R^{lea}_{xi} & N^{bs}_{yi} & W^{dc}_{wi} & R^{dro}_{hi} & \hat{G}_{\theta i} \\
\vdots & \vdots & \vdots & \vdots & \vdots & \vdots & \vdots & \vdots & \vdots \\
N^{ck}_{1n} & S^{ck}_{2n} & N^{nl}_{1n} & F^{ac}_{2n} & R^{lea}_{xn} & N^{bs}_{yn} & W^{dc}_{wn} & R^{dro}_{hn} & \hat{G}_{\theta n}
\end{bmatrix}
\tag{8}
$$

where $N^{ck}$ represents the number of convolution kernels, $S^{ck}$ represents the size of the convolution kernel, $N^{nl}$ represents the number of network layers, $F^{ac}$ represents the activation function, $R^{lea}$ represents the learning rate, $N^{bs}$ represents the number of batch samples, $W^{dc}$ represents the weight decay coefficient, $R^{dro}$ represents the dropout ratio, and $G$ is the regression parameter.

b　Encoding: encoding length when encoding in binary;

$$
L = \log_2 \frac{b-a}{eps} + 1
\tag{9}
$$

where $(a, b)$ is the value range of the independent variable, eps is the required precision, and when there are multiple independent variables, the code length is the sum of the code lengths of each independent variable.

$$L = \sum_{i=1}^{n} L_i \tag{10}$$

c　　Decoding: Convert binary numbers to decimal numbers;

$$x = a + (b - a) \times X / (2^L - 1) \tag{11}$$

where $L$ is the encoding length, $X$ is the binary data.

d　　Fitness function: find the minimum value;

$$F_1(x) = \begin{cases} C_{\text{max}} - f(x) & f(x) < C_{\text{max}} \\ 0 & otherwise \end{cases} \tag{12}$$

where $C_{max}$ is an appropriately large number, and $f(x)$ is determined as the objective function by Formula (4).

e　　Scale transformation of fitness function: here, the dynamic linear transformation method is selected to search for the optimal solution;

$$\begin{aligned} F(x) &= aF_1(x) + b \\ a &= -1, b = F_{1\text{max}} + \xi^k \\ \xi^0 &= M, \xi^k = \xi^{k-1} \times r \\ r &\in [0.9, 0.999] \end{aligned} \tag{13}$$

where $M$ represents the total number of individuals in the population, and $r$ is a random number between [0.9, 0.999].

### 3.5. Visualization of the Optimization Process

When solving the minimum value of the loss function, the genetic algorithm can be used to generate the initial value, and then the gradient descent method can be used to iteratively solve the problem, and finally the global minimum value of the loss function can be obtained (Figure 3). The calculation process of the gradient descent method is essentially to find the minimum value along the gradient descent direction. The iterative formula for gradient descent is Equation (11).

$$x^{(k+1)} = x^{(k)} + \vartheta_k d^{(k)} \tag{14}$$

where $d^{(k)}$ is the search direction starting from $x^{(k)}$, taking the gradient descent direction at point $x^{(k)}$.

$$d^{(k)} = -\nabla f\left(x^{(k)}\right) \tag{15}$$

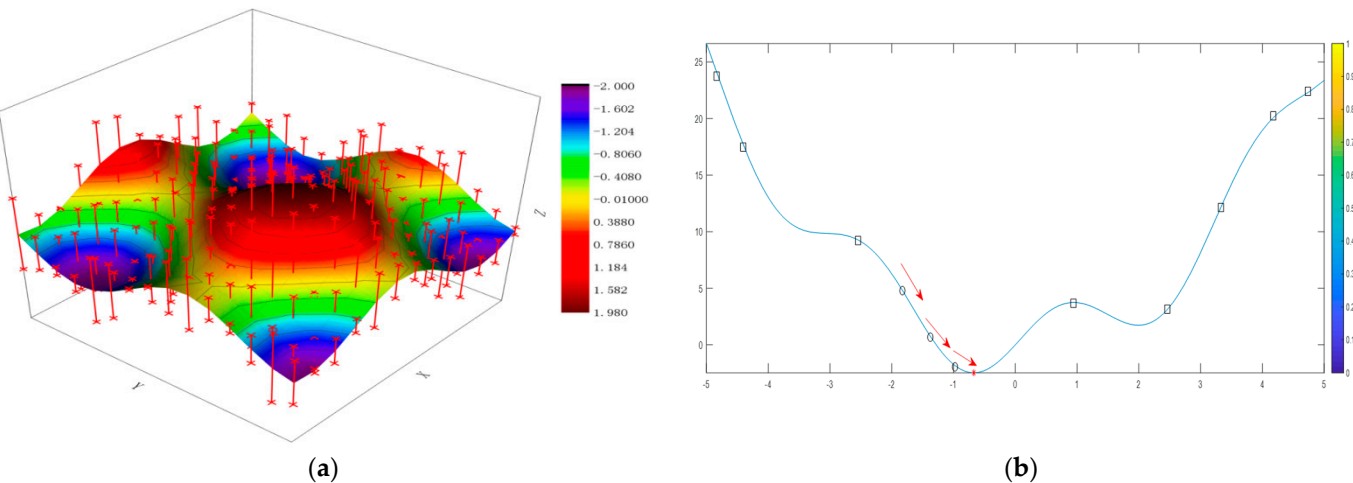

|  (**a**)  |  (**b**)  |

**Figure 3.** Using genetic algorithm combined with gradient descent method to find the optimal solution. (**a**) is the initialization of the population and the Red Cross represents the initial population, and (**b**) looks for the lowest loss function, the rectangles represent the initial population, the circles represent the changes in the population under the rule of gradient descent, and the red arrow represents the path the algorithm takes to find the optimal solution.

$\vartheta_k$ is the step size of one-dimensional search from $x^{(k)}$ along the direction $d^{(k)}$, that is, $\vartheta_k$ satisfies:

$$f\left(x^{(k)} + \vartheta_k d^{(k)}\right) = \min_{\lambda \geq 0} f\left(x^{(k)} + \vartheta_k d^{(k)}\right) \tag{16}$$

To sum up, this paper uses the genetic algorithm to determine the hyperparameters of the neural network classifier, and uses these optimized hyperparameters as the initial population of the genetic algorithm. Each chromosome contains the desired values for the above parameters. This paper updates the definition of the total loss function of target detection, and stipulates that the smaller the loss function, the higher the fitness. After determining the initial parameters, the gradient descent method (GD) is used to find the minimum value and then the global minimum value is obtained. Figure 1 shows the optimization process of the genetic algorithm combined with gradient descent (GA-GD) proposed in this paper. Figure 3a represents the initialization of such a group, and Figure 3b shows the optimization process.

## 4. Experiment

To verify the efficiency of the improved genetic algorithm, we tested it on several complex multimodal functions. Furthermore, to verify the performance of the improved genetic neural network (IGN), we graft the IGN to the classical object recognition algorithm [38]. For example, YOLOv5, Faster RCNN, and SSD. By comparing experiments with commonly used algorithms, the results of the experiments are analyzed qualitatively and quantitatively. The following first introduces the data set and experimental environment used in this paper, then shows the optimization performance of the improved genetic algorithm, and finally analyzes the target detection effect of the genetic neural network.

### 4.1. Experimental Environment

The hardware environment of this experiment is a central processing unit (CPU) Intel i5-7300HQ, and a graphics processing unit (GPU) RTX3080Ti. Memory 128 G. The software environment is Anaconda3, Cuda11.3, Python3.7, Pytorch1.6.0. The entire training process uses a genetic neural network to learn and update the network's hyperparameters. The initial learning rate is 0.03, the initial decay coefficient is 0.0005, the initial batch size is 8, the initial number of training rounds is 200, and the image input size is $640 \times 640$. The

parameters (weights and biases) throughout the training process are learned and updated using stochastic gradient descent.

The dataset used in this paper is collected by the experimental group from the Kaggle (www.kaggle.com, accessed on 19 January 2021) platform. It contains 10,000 datasets covering common animals and plants. These data are divided into two parts, one part is used for the training set (8000 images) and the other part is used for the test set (2000 images). Each piece of data contains at least three different categories of things. We annotated 8000 training sets in Make Sense (www.makesense.ai, accessed on 19 January 2021) software (Version 1.10.0).

### 4.2. Verify the Performance of the Improved Genetic Algorithm

In the process of intelligent optimization algorithm, it is easy to fall into local optimum. To this end, this section will verify the search ability of the improved algorithm in the entire context space. In order to explore the ability of the improved genetic algorithm to find the optimum, 10 complex multimodal functions were selected to test the optimization effect, as shown in Table 1.

**Table 1.** Testing the Improved Genetic Algorithm with Multimodal Functions.

| No. | Function | Range | Range |
|---|---|---|---|
| 1 | $f_1(\mathbf{x}) = \frac{1}{2} \sum\limits_{i=1}^{n} \left( x_i^4 - 16x_i^2 + 5x_i \right)$ | $x_i \in [-5, 5]$ | $-39.166$ |
| 2 | $f_2(\mathbf{x}) = - \sum\limits_{i=1}^{n} \sin(x_i) \sin^{20} \left( \frac{ix_i^2}{\pi} \right)$ | $x_i \in [0, \pi]$ | $-1.801$ |
| 3 | $f_3(\mathbf{x}) = - \cos(x_1) \cos(x_2) \exp \left( -(x_1 - \pi)^2 - (x_2 - \pi)^2 \right)$ | $x_i \in [-100, 100]$ | $-1$ |
| 4 | $f_4(\mathbf{x}) = \sum\limits_{i=1}^{n-1} \left[ 100 \left( x_{i+1} - x_i^2 \right)^2 + (x_i - 1)^2 \right]$ | $x_i \in [-5, 10]$ | $0$ |
| 5 | $f_5(\mathbf{x}) = \left( 4 - 2.1x_1^2 + \frac{x_1^4}{3} \right) x_1^2 + x_1 x_2 + \left( -4 + 4x_2^2 \right) x_2^2$ | $x_1 \in [-3, 3] x_2 \in [-2, 2]$ | $-1.0316$ |
| 6 | $f_6(\mathbf{x}) = \sin(x_1 + x_2) + (x_1 - x_2)^2 - 1.5x_1 + 2.5x_2 + 1$ | $x_1 \in [-1.5, 4] x_2 \in [-3, 4]$ | $-1.913$ |
| 7 | $f_7(\mathbf{x}) = 2x_1^2 - 1.05x_1^4 + \frac{x_1^6}{6} + x_1 x_2 + x_2^2$ | $x_i \in [-5, 5]$ | $0$ |
| 8 | $f_8(\mathbf{x}) = 100\sqrt{\left| x_2 - 0.01x_1^2 \right|} + 0.01 \left| x_1 + 10 \right|$ | $x_1 \in [-15, -5] x_2 \in [-3, 3]$ | $0$ |
| 9 | $f_9(\mathbf{x}) = $ $-0.0001 \left( \left| \sin(x_1)\sin(x_2) \exp \left( \left| 100 - \frac{\sqrt{x_1^2 + x_2^2}}{\pi} \right| \right) \right| + 1 \right)^{0.1}$ | $x_i \in [-10, 10]$ | $-2.0626$ |
| 10 | $f_{10}(\mathbf{x}) = - \frac{1 + \cos\left( 12\sqrt{x_1^2 + x_2^2} \right)}{0.5\left( x_1^2 + x_2^2 \right) + 2}$ | $x_i \in [-5, -5]$ | $-1$ |

Hyperparameters will directly affect the training effect of the model. For example, the number of training epochs is one of the important factors affecting detection performance. If the epoch is too large, the training time will be too long, and even overfitting will occur. If the epoch is too small, the model will not be able to fully learn the feature information of the target, resulting in a low accuracy rate. To better illustrate the impact of hyperparameters on the model, we tested it on CIFAR-10, using epoch as an example to visualize the training process.

Figure 4 shows the GA-YOLOv5 network model proposed in this paper to study the relationship between the number of training epochs and the loss function, accuracy, and time. When the number of training rounds is greater than 10, the decreasing speed of the training loss value and the test loss value begins to become very slow. When the number of training rounds is 18, the recognition accuracy of the model is up to 98.3%. The training time is roughly linear with the number of rounds. The tenth round takes 160 h, the twentieth round takes 260 h, and each additional round takes 10 h on average. It can be seen that it is difficult to give reasonable values for the artificially defined initial hyperparameters, and the hyperparameters should be updated in real time according to the training loss function.

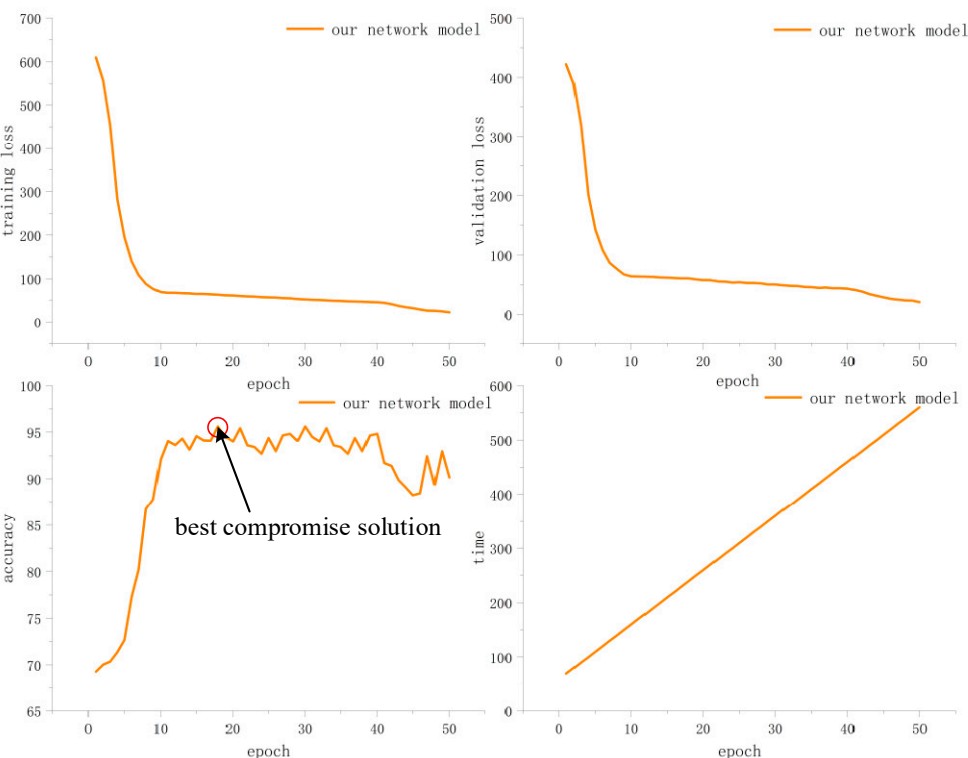

**Figure 4.** Visualize the effect of hyperparameters (epochs) on training results.

### 4.3. Quantitative Analysis of Experimental Results

To verify the performance advantages of genetic neural networks, four sets of experiments were conducted using self-made data. We selected the most popular YOLOv5, Faster RCNN and SSD as the control group. As we all know, *accuracy*, *precision*, *recall*, *IOU*, and *FPS* are commonly used detection indicators for target recognition algorithms [39]. In addition, we also established four parameters *TP*, *FP*, *TN*, *FN* to construct the equation. They are calculated as shown in Equations (14)–(18).

$$Accuracy = \frac{TP}{TP + TN + FP + FN} \tag{17}$$

$$Precision = \frac{TP}{TP + FP} \tag{18}$$

$$Recall = \frac{TP}{TP + FN} \tag{19}$$

$$IOU = \frac{Int}{Uni} \tag{20}$$

$$FPS = \frac{frame}{time} \tag{21}$$

Among them, *TP* means that positive samples are correctly classified as positive samples, *FP* means that negative samples are wrongly classified as positive samples, *TN* means that negative samples are correctly classified as negative samples, and *FN* means that positive samples are wrongly classified as negative samples. Int represents the area of the intersection between the predicted box and the real box, and Uni represents the union area of the predicted box and the real box. Frame represents the number of frames, and time is the time of the test data.

Here, we evaluate the performance of genetic neural networks on classification tasks and plot the IOU–recall and recall–accuracy curves (shown in Figure 5). In the figure, A represents the training curve change for airplanes, B represents the training curve change

for birds, C represents the training results for pedestrians, D is fish, E is car, F is flower, G is insect. In the Figure 5, red represents GA-YOLOv5, black represents YOLOv3, purple represents Faster RCNN, blue represents SSD, green represents YOLOv5, orange represents GA-YOLOv3, and yellow represents GA-SSD.

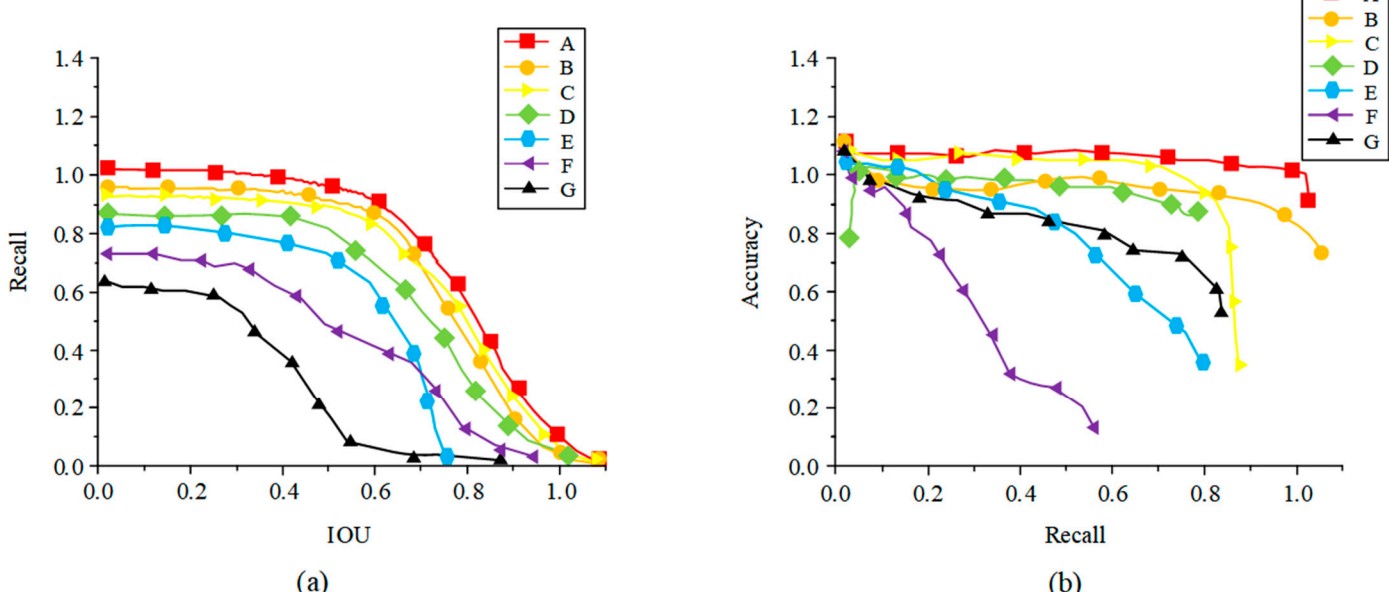

**Figure 5.** Performance comparison of different algorithms in the training phase. (**a**) is the IOU–recall curve and (**b**) is the recall–accuracy curve.

In Figure 5a, when detecting the region of the object, the recall gradually decreases with the increase of the IOU threshold. Among them, the recall rate of SSD algorithm and YOLOv3 algorithm varies greatly with IOU. This shows that the above two algorithms do not perform well in the task of small object recognition. While the GA-SSD and GA-YOLOv3 algorithms have less variation in recall. The recall rates of GA-SSD and GA-YOLOv3 are significantly higher than SSD and YOLOv3 under the same IOU threshold. Compared to other algorithms, the GA-YOLOv5 algorithm reported here achieves the best predictive performance.

According to the PR curve in Figure 5b, the recall decline trend of the Faster RCNN algorithm is faster than other algorithms, indicating that the algorithm has poor positioning of the bounding box. In addition, compared with other algorithms, the GA-YOLOv5 algorithm has the highest accuracy under the same recall. It shows that the target recognition effect of GA-YOLOv5 algorithm is the best. By comparing YOLOv5 and GA-YOLOv5, it is clear that this algorithm is more suitable for determining hyperparameters by genetic algorithm. This further proves the feasibility and rationality of the proposed method.

### 4.4. Qualitative Analysis of Experimental Results

To better demonstrate the performance capability of genetic neural network in object recognition, we show the effect of GA-YOLOv5 model detection in four complex backgrounds. Additionally, small goals should also be included in one of the performance reviews. In Figure 6, the red rectangles identify giraffes, the green rectangles identify hippos, the purple rectangles identify humans, the yellow rectangles identify turtles, the white rectangles identify small fish, the black rectangles identify birds, and the blue rectangles identify camel.

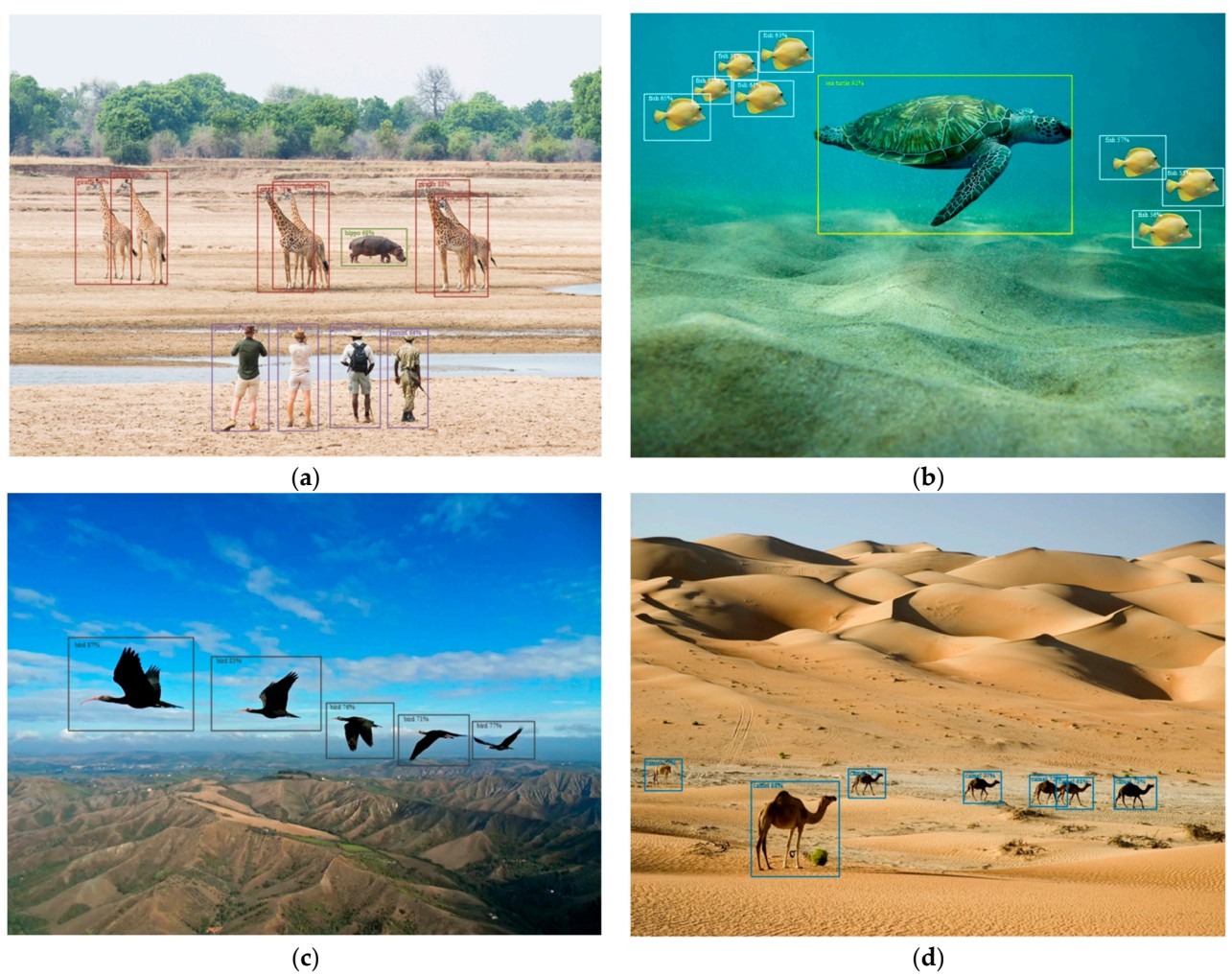

**Figure 6.** Detection effect of GA-YOLOv5 algorithm in four backgrounds. (**a**–**d**) represent the detection results from different backgrounds, a is from the savannah, (**b**) is from the ocean, c is from the sky, and (**d**) is from the desert.

To better illustrate the detection performance of GA-YOLOv5, we show the detection performance of GA-YOLOv5 for small objects in complex backgrounds. In Figure 6a, we choose a figure from the prairie. Among them, the detection accuracy of giraffes is higher, while that of humans is lower. Considering the reason, the characteristic information of giraffes is relatively obvious, while human beings have clothing decoration, and the characteristics are relatively less obvious. In Figure 6b, we choose a figure from the ocean. Among them, the detection accuracy of sea turtle is 92%, while that of fish is lower. Consider that it may be because small fish are small targets and are denser. In Figure 6c, we choose a picture from the sky. Among them, the detection accuracy of the larger birds was up to 87%. In Figure 6d, we choose an image from the desert. This picture well verifies the detection effect of GA-YOLOv5 on small targets. Among them, the detection accuracy of the larger camel was 88%, and the worst detection accuracy was 68%. The detection results under the above four complex backgrounds fully demonstrate the superiority of the GA-YOLOv5 algorithm.

*4.5. Ablation Experiment*

In order to better illustrate the contribution of the optimized hyperparameter method proposed in this paper to the field of object detection, we have selected the best object recognition algorithm at present. For example, Cascade RCNN, Faster RCNN, SSD, YOLOv5, RetinaNet, PV-RCNN. These algorithms include a 2D image detection network and a 3D

point cloud detection network. For small target tasks, whether the improved algorithm can effectively detect has a great relationship with the average accuracy. We choose mean accuracy rate (mAP) as the evaluation metric for ablation experiments. The AP value is obtained by calculating the area under the curve drawn by combining the precision and recall points. Confidence can evaluate the effect of these parameters well, and when the confidence is dense enough, better precision and recall can be obtained. In addition, FPS will also be used to judge the improvement of the detection time brought by the optimized hyperparameters to the model.

This experiment compares and analyzes the test results of the optimized hyperparameters on the classic target detection algorithms YOLOv5, SSD, and Faster RCNN. All experimental environments and computer configurations for this experiment are the same, and the data set used comes from the public cifar-100 data. Table 2 shows that YOLOv5 has a better performance overall, especially with the help of the genetic algorithm, the indicators have been significantly improved, the mAP indicator has increased by 7 percentage points, and the FPS has also increased from the original 15 to 20. It can be seen that the optimized hyperparameters did bring performance improvements to these algorithms. However, there are also situations where the performance of some algorithms is unchanged, such as the IOU indicator in CascadeRCNN and the recall indicator in PV-RCNN. Analyzing the reason, these algorithms themselves include the function of adjusting hyperparameters. Although it is not as comprehensive as ours to adjust hyperparameters, we have to admit that they do play a positive role in some aspects. The overall experiment shows that the optimized hyperparameters can increase the mAP in the target detection algorithm by an average of 3 percentage points, and the FPS can be increased by at least 2 frames. Large-scale commercial application value provides high returns by improving the speed and accuracy of object detection algorithms, which is what the algorithm proposed in this paper is about.

**Table 2.** Comparison of six different detection methods.

| Method | Backbone | Accuracy | Precision | Recall | IOU | mAP | FPS |
|---|---|---|---|---|---|---|---|
| Cascade RCNN | ResNet-101 + FPN | 0.65 | 0.75 | 0.76 | 0.68 | 0.86 | 5 |
| | GA + ResNet-101 + FPN | 0.71 | 0.76 | 0.83 | 0.68 | 0.88 | 7 |
| Retina Net | ResNet-101 + FPN | 0.76 | 0.77 | 0.83 | 0.69 | 0.87 | 7 |
| | GA + ResNet-101 + FPN | 0.84 | 0.78 | 0.83 | 0.71 | 0.91 | 10 |
| PV-RCNN | 3D Voxel | 0.43 | 0.51 | 0.65 | 0.56 | 0.71 | 4 |
| | GA + 3D Voxel | 0.54 | 0.55 | 0.65 | 0.59 | 0.69 | 8 |
| Yolov5 | DarkNet-53 | 0.88 | 0.91 | 0.89 | 0.87 | 0.82 | 15 |
| | GA + DarkNet-53 | 0.94 | 0.95 | 0.96 | 0.94 | 0.89 | 20 |
| SSD | VGG-16 | 0.75 | 0.87 | 0.73 | 0.67 | 0.81 | 15 |
| | GA + VGG-16 | 0.76 | 0.88 | 0.74 | 0.66 | 0.83 | 21 |
| Faster RCNN | ResNet-101 + FPN | 0.79 | 0.68 | 0.73 | 0.81 | 0.87 | 9 |
| | GA + ResNet-101 + FPN | 0.82 | 0.75 | 0.74 | 0.86 | 0.88 | 12 |

## 5. Conclusions

As an important factor affecting the model detection effect, hyperparameters often rely on artificially set initial values. In addition, during the training process, these hyperparameters also need to be updated in real time, which depends on the experience of the algorithm engineer. To this end, this paper creatively proposes to optimize these hyperparameters by using an improved genetic algorithm. First, we propose an improved genetic algorithm and verify the effectiveness of the algorithm in multimodal functions. Then, an improved genetic neural network is proposed; we redefine the loss function of object detection and set it as the objective function of the genetic algorithm. Finally, through qualitative and

quantitative analysis, the superiority of genetic algorithm to optimize hyperparameters is proved.

However, there are many factors that affect the accuracy of object detection. For example, the quality of the dataset, the rationality of the intelligent optimization algorithm, and the complexity of the image background. In the future work, we should also do the following work: (1) Analyze the influence of the calculation amount of the intelligent optimization algorithm on the target detection speed so as to improve the detection speed of the model. (2) Improve the low-quality datasets, such as their exposure and blur functions to improve the detection effect of the model in low-quality images. (3) Increase the number of convolutional layers of the network, fully mine the feature information of different targets, and improve the feature extraction ability of the model.

**Author Contributions:** M.Z. designed the main experiments and completed the writing of the paper; B.L. provided the data set and participated in the annotation of the data set; J.W. provided innovations and constructive comments for the idea of the paper. All authors have read and agreed to the published version of the manuscript.

**Funding:** This research received no external funding.

**Institutional Review Board Statement:** Not applicable.

**Informed Consent Statement:** Not applicable.

**Data Availability Statement:** The data of this study is owned by the research group, and the data sets and codes can be requested from us by peers and by email.

**Conflicts of Interest:** The authors declare no conflict of interest.

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
