# Peer review of "Optimization of Hyperparameters in Object Detection Models Based on Fractal Loss Function"

_fractalfract, doi:10.3390/fractalfract6120706_

Round 1

Reviewer 1 Report

The introduction is a good summary of what is intended, but the first paragraph is disconnected from the second. In the introduction many acronyms are introduced without explanation, the concept of "loss function" is not in the introduction. A more progressive introduction of the relevant concepts is needed. Equation (1) is given without explaining what the terms involved are.

In the body it only mentions once the word fractal in line 317 "fractals and fractions" and the phrase seems decontextualized, it seems to have been inserted to fit with the title of the journal, the term "fractal" should be much better justified in the title and contextualized with the scope of the journal. In addition, there are numerous references that do not seem to have anything to do with the subject of the article (they seem to be references intended to inflate the number of citations of certain authors).

The general idea of looking for a non-manual way to determine the hyperparameters that are used in the interactive neural network is interesting and the implementation that is done seems well thought out and gets good results. That part of the article is interesting, but it also makes few concessions to the general reader, although certainly the general explanation is a little more developed than in the case of INNs.

My recommendation is to rewrite the article with a generalist reader more in mind or looking for a more specific journal for the topic presented, and then you can enter into particular comments on the method, but in the present form of the article it does not make sense to go into more specific details, which would also require discussion or justification.

Author Response

Response to Reviewer 1 Comments

We again thank Reviewer 1 for his valuable comments on our manuscript, which have been of great help in improving the quality of our manuscript. We have re-edited various parts of the manuscript based on the comments of the reviewers. Below I'll give a point-to-point response to each comment.

Point 1:The introduction is a good summary of what is intended, but the first paragraph is disconnected from the second. In the introduction many acronyms are introduced without explanation, the concept of "loss function" is not in the introduction. A more progressive introduction of the relevant concepts is needed. Equation (1) is given without explaining what the terms involved are.

Response 1:Thank you very much for your questions about our manuscript, which will greatly improve the quality of the manuscript. Regarding the problem you mentioned about the disconnection between the first paragraph and the second paragraph in the introduction, we found that there are indeed some problems here. In order to allow readers to better understand the content of the article, we have made the following changes:

Object detection is an applied mathematical technique based on the geometric and statistical characteristics of objects, which plays an important role in various fields [1]. For example, intelligent monitoring, medical navigation surgery, and military target detection. However, many models fail when faced with various complex scenarios and real-time processing of targets [2]. In order to improve the ability of computer vision to cope with complex environments, various scholars have made efforts [3]. The research trends for improving performance can be divided into two directions: one is to improve the ability of model feature extraction from the network framework, such as geometric detectors, attention mechanisms, pyramid networks, etc.; the other is to start from the loss function Used to optimize hyperparameters in the network, such as Bayesian-based hyperparameter optimization and whale optimization algorithm-based hyperparameter optimization.

Taking into account the impact of the backbone network on object detection performance. Hua et al. [4] proposed a matrix-information geometry detector based on Bregman divergence. The author first establishes a positive definite matrix and a clutter covariance matrix for each sample, and then redefines the points on the matrix manifold as the discriminator for signal detection. The final experimental results show that the proposed model is stronger than other detectors.

Regarding the issue you mentioned about abbreviations not being explained, we've troubleshooted these issues and provided explanations for these abbreviations. Loss function as a key term, but it does not appear in the introduction, this is indeed a problem. To this end, we introduce related concepts more gradually in the Introduction. The specific changes are as follows:

Considering the impact of hyperparameters on object detection performance, a large number of scholars optimize these hyperparameters using intelligent optimization algorithms. With the loss function as the objective function, the set of hyperparameter values with the smallest loss will be given as the optimal hyperparameter. Here we emphasize the role of the loss function. Generally speaking, the loss function is used to calculate the gap between the forward calculation result of each iteration of the neural network and the real value, so as to guide the next step of training in the right direction.

In addition, in Equation (1), we also give a detailed explanation for the terms involved:

Among them, loss1 is the defined first-class loss function, which has two parameters ( and class),  represents the probability predicted by the model when dealing with multi-classification problems, i represents the labels of all targets in the multi-classification task, and class represents the class assigned to each codes [0,1,2...] of classes, and j represents the total number of all classified objects.

Point 2:In the body it only mentions once the word fractal in line 317 "fractals and fractions" and the phrase seems decontextualized, it seems to have been inserted to fit with the title of the journal, the term "fractal" should be much better justified in the title and contextualized with the scope of the journal. In addition, there are numerous references that do not seem to have anything to do with the subject of the article (they seem to be references intended to inflate the number of citations of certain authors).

Response 2: Regarding your mention of the words "fractals and fractions" appearing out of context on line 317, we found that there was indeed a problem here and reorganized the speech. Also, regarding your suggestion that "Fractals" is a more reasonable title and associated with the scope of the journal, we agree that this is a good suggestion. Thank you for your comments. Finally, with regard to the fact that many of the references you mentioned do not seem to have any relationship to the topic of the article, we have repeatedly screened for these inappropriate citations and re-added references that are more in line with the topic.

Point 3:The general idea of looking for a non-manual way to determine the hyperparameters that are used in the interactive neural network is interesting and the implementation that is done seems well thought out and gets good results. That part of the article is interesting, but it also makes few concessions to the general reader, although certainly the general explanation is a little more developed than in the case of INNs.

Response 3: We found that when describing the results of the experiment, some concessions were indeed made to ordinary readers. Although the general explanation is more complete than the case of INN, more consideration should be given to the reader's understanding of the article. To this end, we supplemented related experiments. To better illustrate the impact of optimizing hyperparameters with intelligent algorithms on model performance, we performed ablation experiments. And use multiple indicators to judge the performance of the model.

Point 4:My recommendation is to rewrite the article with a generalist reader more in mind or looking for a more specific journal for the topic presented, and then you can enter into particular comments on the method, but in the present form of the article it does not make sense to go into more specific details, which would also require discussion or justification.

Response 4: Regarding your question about the correlation between the form of the article and the journal, we first emphasized the correlation between the content of the article and the fractal, and then pointed out that the purpose of this article is to construct a fractal loss function. It is worth emphasizing that the classification loss function here is different from the existing object detection loss function.

As we all know, "Fractal" was originally a geometric concept, meaning multi-dimensional, super-dimensional or fractal-dimensional. It broke the original concept of three-dimensional space, and it can not only exceed three-dimensional but also have fractional dimensions. In fact, whether it is to verify the optimization process of the improved genetic algorithm, or to optimize the hyperparameters with the loss function as the goal, they are all carried out in a multi-dimensional space. In addition, in the process of optimizing the model, it is itself in a multi-dimensional space, using points, lines, surfaces, and even space-time to describe the trajectory of the algorithm. "Fractal" can not only be used to describe objects that exist in nature, such as tortuous and irregular lightning paths or curved and complex coastal shapes, but it can also describe the optimization path of algorithms.

In addition, in the process of finding the optimal parameters with the fractal loss function as the goal, the trajectory of the algorithm should belong to the category of fractal theory. The entire optimization process adopts the method of stochastic gradient descent. We fully understand the law of gradient descent and devote ourselves to a shorter path to obtain the global optimal solution. We read a lot of published articles from "Fractal Fract", they include the study of the shape of the strait or the use of fractal theory to achieve complex curve fitting techniques. We consider the methodology in this paper to be of relevance to journals. However, we acknowledge that these related content seem to be obscured by the application of object detection. To this end, we have revised the content of the article and made a more specific explanation for fractal functions.

Reviewer 2 Report

Dear Editor,

As requested I have reviewed the article, as mentioned above, by Ming Zhou, Bo Li and Jue Wang. I enumerate my observations in the following.

· The authors propose, as title of the paper suggests, optimization based strategy in selecting hyperparameters in object detection exercise using neural network (NN). The techniques of object detection encompass a large class of subjects. The statement made by the authors “Object detection is an applied mathematical technique based on geometrical and statical characteristics of objects” in p. 1, line# 27-28 is not quite right. Physical attributes associated with various physical phenomena based techniques, for example, electromagnetic, acoustic, optical with scattering, emission and absorption also play key roles. The starting sentence in the “Introduction” requires rephrasing, so that it should not confuse a reader.

· The title of the paper suggests that the major focus of the paper is to determine an optimal set of hyperparameters associated with object detection where the authors use fractal based loss function in the realm of NN. Actually, the authors have mentioned in the abstract that they used complex fractal function. Unfortunately, the authors remain non-categorical about the type of fractal function used. Since fractal function is an important element of the paper which is categorically defined in the title, it is imperative that the authors specifically devote a section on fractal function associated with the type of target detection. They also need to state why they considered the particular fractal function.

· The authors also need to categorical about the detection of static or moving objects, as with a moving objects the five hyperparameters (two for object coordinates, two for objects and one for number of objects) as mentioned in equation (4) of the loss function are not enough, unless the moving object is treated as pseudo-stationary.

· The model complexity depends largely on the number of hyperparameters, so is the case with a loss function. The loss function is nonlinear and multimodal. The multimodality of the loss function becomes more complex with the increase in the number of hyperparameters. Although the authors have used genetic algorithm, a global optimization algorithm, it is however not clear how the authors effectively overcome the multimodal conditions. I think the authors need to be explicit and categorical about the issue of multimodality.

· The authors, on the other hand, have given a list of test functions with two variables for testing their genetic algorithms. The genetic algorithm (although the authors have proposed modification in it) has been well established global optimization technique which has been tested with all kinds of test functions as presented to overcome multimodality, so much as it has become a plethora. Therefore, it does not carry any special meaning to the proposed paper. Moreover, the number of hyperparameters in their problem is more than two.

· The objective, and truly speaking a real success, of the proposed technique is its applicability of the method of object detection in a real time scenario. In fact, in the ‘Introduction’ section the authors have mentioned “It is understood that the hyperparameters required in the detection task should be updated in real time to improve the effect of model detection” (p.2, line# 70-71). But the results demonstrated by the authors are far from such reality. For example, recognition accuracy depends on training round. To get an accuracy better than 95% with GA-YOLOv5 NN requires 260 hours for 20 rounds training which is equivalent 10.8 days. For 10 rounds of training it takes almost a week of NN training for a GPU based multi-processors system. Note that the curves with variables Recall vs. IOU and Accuracy vs. Recall associated with detecting airplanes and birds are very close even after sufficient training which cast a doubt of practical application of the technique of target detection even with the detection speed of 20 frames per second.

· The authors have mentioned in the ‘Abstract’ that they have used fractal based loss function and fractional calculus, but nowhere in the text they have specified using typical fractal function and fractional calculus. The loss function as defined in equation is generic and it does not indicate the use of any particular fractal function. It is also not clear about the application of fractional calculus.

· The loss function in equation (4) comprises of 5 components, in that two are reserved for coordinates, two for target and one number of targets. All three components are connected through weighting coefficients, such as \lambda_{coord}, \lambda_{coord}, \lambda_{obj}, \lambda_{obj} and \lambda_{nobj}. Again in gradient descent algorithm \lambda_{k}. Evidently, these are different. Therefore, appropriate application of symbols is necessary for clarity.

· The authors have failed to explain symbols used such as the variables with crown are not described at all.  

My critical observation suggests that the paper requires major revisions before it is considered for publication.

Author Response

Response to Reviewer 2 Comments

We are very grateful to the editors and reviewers for your valuable comments on our manuscript, which has greatly helped to improve the quality of our manuscript. We re-edited various parts of the manuscript based on the reviewer’s comments. Below I will give a point-to-point response to each review.

Point 1: The authors propose, as title of the paper suggests, optimization based strategy in selecting hyperparameters in object detection exercise using neural network (NN). The techniques of object detection encompass a large class of subjects. The statement made by the authors “Object detection is an applied mathematical technique based on geometrical and statical characteristics of objects” in p. 1, line# 27-28 is not quite right. Physical attributes associated with various physical phenomena based techniques, for example, electromagnetic, acoustic, optical with scattering, emission and absorption also play key roles. The starting sentence in the “Introduction” requires rephrasing, so that it should not confuse a reader.

Response 1:We very much appreciate your pointing out issues in the manuscript which are of great value. In order to give interested readers a deeper understanding of the content of the article, we have revised the content of this section in the Introduction. The specific changes are as follows:

Object detection is an applied mathematical technique based on the geometric and statistical characteristics of objects [1]. Physical properties related to various physical phenomena based on technologies also play a key role in object detection, such as electromagnetics, acoustics, optics with scattering, emission, and absorption. In recent years, with the continuous development of science and technology, target detection has played an important role in various fields, such as intelligent monitoring, medical navigation surgery, military target detection, etc. However, many models fail when faced with various complex scenarios and real-time processing of targets [2]. In order to improve the ability of computer vision to cope with complex environments, various scholars have made efforts [3]. The research trends for improving performance can be divided into two directions: one is to improve the ability of model feature extraction from the network framework, such as geometric detectors, attention mechanisms, pyramid networks, etc.; the other is to start from the loss function Used to optimize hyperparameters in the network, such as Bayesian-based hyperparameter optimization and whale optimization algorithm-based hyperparameter optimization.

Point 2: The title of the paper suggests that the major focus of the paper is to determine an optimal set of hyperparameters associated with object detection where the authors use fractal based loss function in the realm of NN. Actually, the authors have mentioned in the abstract that they used complex fractal function. Unfortunately, the authors remain non-categorical about the type of fractal function used. Since fractal function is an important element of the paper which is categorically defined in the title, it is imperative that the authors specifically devote a section on fractal function associated with the type of target detection. They also need to state why they considered the particular fractal function.

Response 2:Thank you very much for your question and we agree with you. Since the paper introduces a fractal-based loss function, a section must be dedicated to introduce fractal functions. To this end, we have added this section on fractal functions to the 3.3 section of the article. The specific changes are as follows:

Consider that the classification loss is often directly related to how well the model predicts. If the classification loss is too large, it indicates that the model cannot accurately extract the features of the image. It even leads to confusion among similar categories, which is a fatal effect in the object recognition process [34]. However, most of the existing algorithms fail to satisfy the more precise definition of loss function. To solve the above problem, we redefine the total loss function. We divide the loss function into a regression box loss function, a confidence loss function, and a classification loss function. The overall loss function is shown in Equation 4.

Here, coord indicates that the input layer has additional coordinate information channels, P indicates the predicted classification probability, i and j indicate coordinate information, (x, y) indicates the center coordinates of the rectangular box, w and h indicate the width and height of the rectangular box, C stands for confidence, M stands for the number of categories, K stands for all clusters, λ stands for the parameters used to balance these loss functions, c stands for the class of classification, and those symbols with crowns represent the true value.

As a loss function to evaluate the gap between prediction and reality, it needs to be judged by multi-angle information, and these observation angles also have self-similarity. Considering the multifractal nature of the loss function, we analyze and evolve a system consisting of many interacting elements in space and time. First, assuming that the entire agent population is distributed in a regular space, a representation of the spatial coordinate (x, y) at time t is sought. Time t is used to describe the number of training rounds, and (X, Y) is used to describe the relevant data of the predicted value. In this paper, the box counting method is used to estimate the fractal dimension of an object as the slope of a linear fit between the logarithm of the number of boxes required to optimally cover the object and the logarithm of the box size. We follow Wang et al. [35] and consider defining the fractal loss function in terms of the integral and its continuous limit time derivative, as shown in the following equation.

Here, x represents the center coordinates of the prediction frame, y represents the size of the prediction frame, R represents an arbitrary function (generally refers to weighing the importance of each parameter by establishing weights), and t represents time (number of rounds). Assume that the evolution of f(x,y,t) is fractal in time, and express the fractal by fractional derivatives of order β. Using the Bayesian variation, the expression in Equation 6 is updated as:

 Eversing the order of integration and using the fractional Taylor expansion R(x,y) is expressed as:

where γ is the fractional order related to the fractal structure of agent evolution, and (x, y) represents the center coordinates of the training box in the first round.

Point 3:  The authors also need to categorical about the detection of static or moving objects, as with a moving objects the five hyperparameters (two for object coordinates, two for objects and one for number of objects) as mentioned in equation (4) of the loss function are not enough, unless the moving object is treated as pseudo-stationary.

Response 3:Dear reviewer, regarding the analysis of moving object detection that you mentioned, I have to say that your proposal is very valuable, and this will be a direction in our future work. However, even the detection of moving objects at this stage is essentially composed of frames of still images. As long as the processing time of the algorithm is too fast, there will be no problem of frame loss. We believe that as long as stationary objects are handled well, coupled with fast processing power, dynamic object detection should not be a problem. In addition, we also tested the performance of the loss function in dynamic videos, and the results showed that the average detection accuracy was 92.1% based on the FPS of 25. I'm not sure if I can upload motion video while replying to your comments, so uploading it to https://github.com/Smart-Zhou369/Target-Detection.

Point 4:4) The model complexity depends largely on the number of hyperparameters, so is the case with a loss function. The loss function is nonlinear and multimodal. The multimodality of the loss function becomes more complex with the increase in the number of hyperparameters. Although the authors have used genetic algorithm, a global optimization algorithm, it is however not clear how the authors effectively overcome the multimodal conditions. I think the authors need to be explicit and categorical about the issue of multimodality.

Response 4:Dear reviewer, I have to say that the question you raised is still very deep. Regarding the question you raised, I will answer it in two parts: (1) About how to solve the multimodal problem. (2) How to overcome the multimodal problem.

As the number of hyperparameters increases, the multimodality of the loss function also becomes complex. When faced with a multimodal problem, our solution is to transform it into a single-modal problem that is easier to solve. According to the different principles of target conversion, we adopt a more reasonable hierarchical sequence method. The basic idea is: sort the multi-objectives according to their importance, first find the optimal solution of the first goal, and then find the optimal solution of the second goal under the condition of reaching this goal, and so on until the last solution At the end, the multimodal optimality is obtained. Of course, this scheme is not necessarily the best, but it is easy to understand and basically meets the requirements when solving hyperparameter optimization problems. Specifically, it is shown in Equation 1. Considering the logic of the article, we believe that this part should not be the focus, and it should not appear in the main text.

As for how to overcome the problem of multi-peak conditions, its essence is to study the relationship between local optimum and global optimum, which is exactly the improvement of genetic algorithm discussed in this paper. The traditional method uses elite reservation or roulette as the selection operator. The disadvantage of the former is that it is easy to fall into a local optimum, and the disadvantage of the latter is that it cannot retain the best individuals well. When facing the multimodal problem, the first solution in this paper is to start from the selection operator. We propose a multi-objective genetic algorithm based on ranking selection, which directly copies the 5% individuals with higher fitness to the next generation. In addition, 10% of individuals with high fitness will replace individuals with low fitness, and the newly formed community will complete crossover and mutation. Although the optimal individual participates in the crossover and mutation operations, it is easy to fall into local optimum by directly copying the top 5% of the optimal individual to the next generation each time. In order to get the global optimal solution, we increase the mutation rate from 0.03 to 0.1. If there are still multiple peaks in the result of such an operation, then there is always one value that is the largest.

Point 5:· The authors, on the other hand, have given a list of test functions with two variables for testing their genetic algorithms. The genetic algorithm (although the authors have proposed modification in it) has been well established global optimization technique which has been tested with all kinds of test functions as presented to overcome multimodality, so much as it has become a plethora. Therefore, it does not carry any special meaning to the proposed paper. Moreover, the number of hyperparameters in their problem is more than two.

Response 5:Dear reviewer, I think the questions you raised are enlightening for us. We admit that a large number of existing intelligent optimization algorithms have good performance results in solving multi-modal problems, even better than the results presented by the improved genetic algorithm in this paper. However, our proposed improved algorithm is dedicated to solving practical problems. In fact, when solving hyperparameter optimization problems, it does not mean that a certain algorithm can get a more precise optimal value and it is good. It should be emphasized that we hope that each round of training model will automatically output a set of hyperparameters based on the loss value of this round. However, this in itself imposes a huge computational burden on the model when calculating the difference between the true and predicted values. Therefore, it is no longer possible to increase the burden on the network in order to pursue the optimal parameters. In fact, a simple principle plus a simple algorithm is enough to solve the optimization problem of hyperparameters. The improvement principle of the genetic algorithm in this paper is simple and easy to understand, but whether it is feasible or not, I think it is necessary to test it through the multimodal function. On the one hand, these tests are to verify the feasibility of the method, and on the other hand, they help readers understand multimodal issues.

Point 6:  The objective, and truly speaking a real success, of the proposed technique is its applicability of the method of object detection in a real time scenario. In fact, in the ‘Introduction’ section the authors have mentioned “It is understood that the hyperparameters required in the detection task should be updated in real time to improve the effect of model detection” (p.2, line# 70-71). But the results demonstrated by the authors are far from such reality. For example, recognition accuracy depends on training round. To get an accuracy better than 95% with GA-YOLOv5 NN requires 260 hours for 20 rounds training which is equivalent 10.8 days. For 10 rounds of training it takes almost a week of NN training for a GPU based multi-processors system. Note that the curves with variables Recall vs. IOU and Accuracy vs. Recall associated with detecting airplanes and birds are very close even after sufficient training which cast a doubt of practical application of the technique of target detection even with the detection speed of 20 frames per second.

Response 6:Dear reviewer, I have to say that your questions are so professional, which is necessary to improve the quality of the article. Below I will answer these questions one by one. First of all, in the "Introduction" section, we have stated that "the hyperparameters required in the detection task should be updated in real time to improve the model detection effect." The results on line 375 are a good proof of the need for real-time update of the hyperparameters. The overall results well reflect the impact of epoch on loss, accuracy and time during training. In fact, the "best compromise solution" in the figure represents the optimal solution (epoch=19), so the model should terminate the training process before the 20th round. However, to illustrate the relationship between hyperparameters and relevant metrics, a full training experiment (epoch=50) was performed. It is worth noting that we update the hyperparameters in real time, so the training has been stopped by the 20th round. On the other hand, regarding the problem you mentioned, the curves of Recall vs. IOU are very close to the variables related to the detection of aircraft and birds. As we all know, the image features of small target aircraft and birds, especially when they are in flight, do have great similarities, so it is not difficult to explain the existence of similar curves. However, it is indeed controversial to deny the innovative type of the article based on the similarity of the curve results. But we also admit the problem of insufficient experiments. In order to better illustrate the contribution of the ideas in this paper to object detection, we supplement ablation experiments.

Point 7:  The authors have mentioned in the ‘Abstract’ that they have used fractal based loss function and fractional calculus, but nowhere in the text they have specified using typical fractal function and fractional calculus. The loss function as defined in equation is generic and it does not indicate the use of any particular fractal function. It is also not clear about the application of fractional calculus.

Response 7:Thank you very much for pointing out the problem of the article, we admit that there is indeed no fractal loss function that is not explicitly proposed here. For this reason, we believe that relevant content should be supplemented in a separate section.

Point 8:  The loss function in equation (4) comprises of 5 components, in that two are reserved for coordinates, two for target and one number of targets. All three components are connected through weighting coefficients, such as \lambda_{coord}, \lambda_{coord}, \lambda_{obj}, \lambda_{obj} and \lambda_{nobj}. Again in gradient descent algorithm \lambda_{k}. Evidently, these are different. Therefore, appropriate application of symbols is necessary for clarity.

Response 8:Dear reviewer, Thank you very much for pointing out the issues in the manuscript. These questions are indeed confusing with regard to the frequent use of lambdas in equations. As you mentioned, we have modified the questions for clarity by adopting the appropriate applied notation.

Point 9:  The authors have failed to explain symbols used such as the variables with crown are not described at all.

Response 9:Thank you very much for pointing out the details in the manuscript, we admit that we did omit the explanation of some symbols, so we have carefully checked such problems and added relevant explanations.

Reviewer 3 Report

Object detection is an applied mathematical technique based on the geometric and statistical characteristics of objects, which plays an important role in various fields such as intelligent monitoring, medical navigation surgery, and military target detection. 

The aim of this paper is to creatively proposes to optimize these hyperparameters by using an improved genetic algorithm.  The authors in this work propose an improved genetic algorithm and verify the effectiveness of the algorithm in multimodal functions. Then, an improved genetic neural network is proposed, we redefine the loss function of object detection and set it as the objective function of the genetic algorithm. Finally, through qualitative and quantitative analysis, the superiority of genetic algorithms to optimize hyperparameters is proved.  Also, in this paper, some interesting perspectives are presented.

I like this idea and believe that the results are suitable for your journal. I recommend the publication of the paper in the journal "Fractal and Fractional".

Author Response

Response to Reviewer 3 Comments

Dear reviewer, Thank you very much for your recognition of our work, which will be something we will continue to work on. My sincerest thanks to you and wish you success in your work.

Round 2

Reviewer 1 Report

I consider that the explanations now adequately contextualize the work and allow a general reader to understand the background of the issue. I also consider the level of rigor and the choice of methods to be pertinent, and the paper makes an interesting and innovative proposal, which deserves to be published.

I still have my doubts about Fractal an Fractional being the most appropriate journal for this work, but in any case, I have no serious objection to the work, which seems to me to be meritorious and self-explanatory.

Reviewer 2 Report

I went through the response by the authors on my reviewed points of discussion, and I am satisfied with the response by the authors.